# Knowledge Translation of Healthcare Research in Saudi Arabia—Implications for Community Health and Primary Care Under the New Saudi Model of Care: A Narrative Review

**DOI:** 10.3390/healthcare13192469

**Published:** 2025-09-29

**Authors:** Ibrahim M. Gosadi

**Affiliations:** Department of Family and Community Medicine, Faculty of Medicine, Jazan University, Jazan 45142, Saudi Arabia; gossady@hotmail.com

**Keywords:** knowledge translation, community health, primary care, evidence-based practice, Saudi Arabia

## Abstract

Knowledge translation (KT) is an essential process in bridging the gap between research evidence and healthcare practice, particularly in community health and primary care settings. In Saudi Arabia, KT is gaining increasing importance as the healthcare system undergoes a major transformation under Saudi Vision 2030 and the new Saudi Model of Care. The new model of care emphasizes the importance of healthy communities and primary care as early elements of healthcare service delivery before reaching the secondary and tertiary healthcare levels. Additionally, healthcare transformation under Saudi Vision 2030 encourages the utilization of evidence and KT to improve healthcare services provided to individuals and enhance the standardization of healthcare delivery. Nonetheless, the application of KT principles in community health and primary care contexts has faced some challenges during the period preceding the establishment of the new Saudi Model of Care. While Saudi Arabia has achieved significant advances in health research and institutional capacity building, KT remains underutilized in local community health initiatives. This narrative review aims to provide a conceptual overview of KT and explore its implications within the contexts of community health and primary care in Saudi Arabia. Additionally, the review introduces the key components of KT—evidence synthesis, dissemination, exchange, and application—and examines how these can be implemented in community and primary care contexts. The review emphasizes the necessity of stakeholder engagement, community-based participatory research, and the integration of frameworks such as the knowledge to action and social–ecological models to ensure effective KT in community health settings. Future directions should focus on expanding KT training, promoting its integration across health institutions involved in primary care delivery, and sustaining community health through strong partnerships among academic, governmental, and community stakeholders.

## 1. Introduction

Knowledge translation (KT) in healthcare has been gaining increasing attention in recent years due to its role in promoting the effective utilization of health research findings to improve healthcare service delivery. According to the Canadian Institutes of Health Research, KT is defined as “a dynamic and iterative process that includes the synthesis, dissemination, exchange, and ethically sound application of knowledge to improve health, provide more effective health services and products, and strengthen the health care system” [1]. Based on this definition, KT comprises four main elements: the synthesis of evidence from healthcare research, the dissemination and communication of this evidence, the bidirectional exchange of knowledge between researchers and knowledge users, and the effective and ethical application of knowledge.

Knowledge translation plays a vital role in community health and primary healthcare settings. Community health refers to the health status of members within a community, the problems affecting their health, and the nature of the healthcare services provided to the community [2]. The health of a community is shaped by demographic, environmental, and socioeconomic factors, along with the health needs of its population and the accessibility and utilization of healthcare services [3]. Primary healthcare serves as a cornerstone in ensuring community health and is regarded as an integral component in the delivery of community-based healthcare services [4].

The evolving nature of communities, with continuously changing demographics, health statuses, and needs and changes in trends pertaining to the determinants of health, indicates the importance of performing the continuous and systematic monitoring of health and its determinants at the community level. Furthermore, how members of communities perceive and utilize healthcare services and how this overall process impacts community health outcomes necessitate the application of KT principles in the synthesis, dissemination, exchange, and application of knowledge. The effective application of a community-based intervention requires the embracement of KT principles when assessing baseline characteristics, selecting priorities pertaining to community health, designing suitable interventions, implementing an intervention and assessing its outcomes, disseminating and communicating the generated knowledge, and cyclically applying the KT process.

In Saudi Arabia, KT is gaining increasing importance as the healthcare system undergoes a major transformation under Saudi Vision 2030. The vision provides a comprehensive roadmap leading to the revolutionary transformation of healthcare in Saudi Arabia. The Health Sector Transformation Program is one of the vision realization programs, aiming to enhance access to healthcare in a manner that ensures meeting the health needs of every individual in the community [5]. The program emphasizes the importance of public health and disease prevention to ensure the health and productivity of individuals and communities.

One of the main health transformation themes introduced by the Saudi Ministry of Health is the New Model of Care. The newly introduced model, recently relabeled as the Saudi Model of Care, focuses on enhancing personal value during the healthcare-seeking process [6]. Earlier reports indicated that accessibility to healthcare in Saudi Arabia was facing challenges, especially with the limited impact of primary healthcare services and growing burden of chronic diseases [7]. The transformation of the healthcare system under Saudi Vision 2030 emphasizes the importance of responding to the health needs of the population and the adoption of a preventive approach, instead of the previously adopted curative approach [8]. Additionally, the transformation includes a shift from independent healthcare service provision toward integrated healthcare services across all levels of healthcare services, including private healthcare services, while adopting the use of electronic healthcare systems [9,10].

This narrative review aims to provide an introduction to KT and its correlation with community health and primary care within the transformed Saudi Arabian healthcare context. The Saudi Model of Care and its emphasis on the importance of utilizing an evidence-based approach toward enhancing the health of individuals and communities necessitate the successful adoption of KT in the community health and primary care contexts. The Saudi Arabian healthcare system has been successful in implementing nationwide public health programs, such as the Healthy Marriage Program and the National Newborn Screening Program [11], as well as the digital transformation of primary care services and the use of telemedicine in the country [12]. Nonetheless, the application of KT principles in community health and primary care contexts has faced some challenges during the period preceding the establishment of the new Saudi Model of Care.

## 2. Methodology

This review briefly presents the core concepts of KT and explains its importance in community health and primary care. Major players in health research and evidence-based practice in the country are outlined due to their relevance to KT. Finally, the review aims to offer a practical application of KT in the context of Saudi Arabian community health and primary care under the new Saudi Model of Care.

To conduct this review, multiple steps were taken to gather content related to the current status of KT in Saudi Arabia and its connection to community health and primary care. First, the relevant literature explaining the principles of KT was consulted. This was followed by a review of the literature concerning community health and primary care and how these areas intersect with KT principles. Second, an overview was developed to present the current status of KT in Saudi Arabia, identifying the main institutions responsible for legislating and implementing knowledge creation and translation, by searching all relevant research-performing entities in the country. Third, studies assessing evidence-based practice and KT in community health and primary care in Saudi Arabia were reviewed.

These steps were carried out by searching Web of Science, PubMed, Google Scholar, and region-specific literature databases such as the Saudi Digital Library. The utilized keywords and MeSH terms were related to knowledge translation, evidence-based practice, the Saudi Model of Care, community health, and primary care. This involved the use of specific terms and keywords related to these main concepts, such as community care, community health services, community medicine, primary healthcare, access to primary care, knowledge translation, translational medical sciences, evidence-based practice, evidence-based healthcare, and healthcare reform.

To enhance the ability to identify literature relevant to the review’s aims, an advanced search strategy to identify published peer-reviewed articles involved the use of Boolean operators (such as AND, OR), including the selected terms in the titles or abstracts of identified articles. No timeframe restriction was applied for the searched literature. Additionally, a grey literature search involved targeting recognized resources and authorities associated with KT in healthcare, KT within community and primary care, evidence-based practice in healthcare settings in Saudi Arabia, and the new Saudi Model of Care. Finally, hand searching of the reference lists of included articles was utilized to identify other potentially relevant literature.

The inclusion and exclusion criteria for the identified literature were based on its relevance to KT and its application within the community and primary healthcare contexts. Articles were included if they were published in the English language, presenting original research, systematic or narrative reviews, statements from recognized establishments and official institutions pertaining to KT, evidence-based practice, the Saudi Model of Care, community health, and primary care. Exclusion of the identified literature was performed if the topics were not relevant to KT within community healthcare and primary healthcare or if not related to the transformation of healthcare within the Saudi Arabian context.

The screening of the identified articles was conducted by a single investigator (I.G.), and no assessment of the risk of bias in the included studies was performed due to the narrative nature of the current review. An effort was made to ensure the selection of relevant publications by screening the titles and abstracts and the application of the inclusion and exclusion criteria. This process aided in the selection of articles relevant to KT, its application within primary care and community health, research and evidence-based practice in healthcare settings in Saudi Arabia, and the new Saudi Model of Care. A list of articles used to generate the findings of the current review is illustrated in the Appendix A.

## 3. Results

### 3.1. Knowledge Translation in Healthcare

Knowledge translation has emerged as a concept in the past three decades, with the overall aim of facilitating the integration of knowledge creation with knowledge application [13,14]. This integration promotes timely connections between researchers who create knowledge and the users of this knowledge. One of the leading institutions to embrace KT was the Canadian Institute of Health Research [1]. In 2005, the World Health Organization began actively promoting KT by establishing the Evidence-Informed Policy Network [15]. Other international institutions promoting KT in healthcare include the National Institute for Health and Care Excellence in the United Kingdom [16], the National Institutes of Health in the United States [17], and the National Health and Medical Research Council in Australia [18].

The global interest in KT in healthcare settings stems from the need to transform health research into clinical practice. However, with the growing volume of published health research, translating this research into practice is becoming increasingly complex. Challenges include the quality and practicality of the synthesized evidence, as well as limitations in the dissemination process. Moreover, it has been reported that the time from the publication of research findings to their practical application can take up to 17 years, indicating a significant delay in implementing research outcomes [19].

The difficulty of applying knowledge generated from research in healthcare settings is due to several factors. However, one of the main reasons for delays in research translation is the limited integration between KT and research planning [20]. A key manifestation of this limited integration is the predominance of research-driven projects rather than user-involved projects. When knowledge users are not actively engaged in the process of creating knowledge, the resulting outputs may see limited adoption in subsequent phases.

Conventional research projects are often based on knowledge gaps identified by researchers in academic institutions. In such cases, researchers may apply scientific methods to answer a specific research question, generating knowledge that fills the gap for academic or publication purposes. While this approach may be welcomed within the scientific community, the resulting knowledge may have limited applicability. This is primarily because it does not involve knowledge users in the processes of conceptualization, enhancing the feasibility and generalizability of the methodology, and facilitating effective dissemination. Multidisciplinary collaboration between research institutions, healthcare providers, and the community is essential to ensure that KT is effectively embedded throughout the research process [21,22].

The shift from researcher-driven research projects to user-involved projects can have several beneficial implications. Involving knowledge users from different disciplines in the conceptualization of research ideas ensures that the research is based on actual health needs and facilitates the selection of relevant priorities. Involving knowledge users in the planning phase can enhance the feasibility, cost-effectiveness, and cultural acceptability of the generated evidence. Subsequently, this integration between researchers and knowledge users can result in a scientifically sound methodology, producing knowledge that is suitable for application in relevant healthcare settings.

Knowledge translation, as a methodology, includes multiple tools and frameworks. The tools for KT vary according to their purpose, such as planning, evidence synthesis, clinical decision support, and dissemination and communication [13,23,24]. These tools aid knowledge translators in navigating multiple steps of the KT process, including creating an integration roadmap, setting goals and outcomes, addressing facilitators and challenges, selecting a KT framework, synthesizing evidence, mobilizing and disseminating knowledge, implementing strategies, evaluating outcomes, and planning for sustainability.

It must be noted that these steps are not arranged in a linear pathway but can be multidirectional. For example, designing a clear roadmap for integration can facilitate implementation. Early mobilization of the KT plan can help in setting realistic and practical goals. Finally, evaluating the KT plan can lead to the generation of new knowledge, further enhancing future evidence synthesis. Unlike research planning from the sole viewpoint of the researcher, the steps of KT are often interconnected and complex. Acknowledging this complexity can contribute to the success of the KT process.

### 3.2. Intertwining of Community Health, Primary Care, and Knowledge Translation

The health of individuals is affected by their surrounding environments and their communities. Healthy communities ensure the well-being of residents, while unhealthy communities increase the risk of chronic diseases [25]. A healthy community is shaped by several determinants, including social, economic, and environmental components [26]. These determinants are associated with the exposure of individuals to risk factors, requiring continuous and periodic assessments and updates.

One of the methods for diagnosing a community’s health status is a community health needs assessment. The assessment of community health needs has three main components: measuring the health status of community members, assessing the exposure to risk factors at the community level, and identifying actions and interventions required to address these issues [1]. The success of applied interventions is monitored and evaluated to assess their effectiveness. Additionally, the process of assessing community health needs, deciding priorities, and selecting community-based interventions requires collaborative multidisciplinary actions.

The nature of community health and primary care as medical disciplines necessitates the successful application of the KT process dynamically and cyclically. Figure 1 displays a roadmap for planning KT in a community health context. The roadmap progresses through multiple phases, starting with planning integration between knowledge translators and knowledge users; setting community-oriented goals; mitigating facilitators and challenges relevant to the community context; selecting a framework that combines community-based participatory research (CBPR), the socioecological model (SEM), and the knowledge to action (KTA) framework; synthesizing evidence suitable for the local community context and healthcare system; planning the dissemination of synthesized knowledge; planning integrative implementation; evaluating the applied knowledge; and assessing sustainability. Furthermore, these phases can be flexible and dynamic, where findings obtained in one phase can be used to modify or enhance the KT process accordingly.

Successful KT in community health requires effective and early integration between the KT teams and knowledge users. Limited integration can be a barrier to KT due to differences in goals and interests between those involved in knowledge creation and those using the knowledge in community settings [26]. The KT team should be multidisciplinary, equipped with the necessary skills, and supported logistically and legislatively to facilitate the effective application of the methodology required to implement the synthesized knowledge. Early integration should ensure the involvement of the KT team with the relevant stakeholders associated with community health from the outset.

Stakeholder involvement can be mapped according to their potential contribution, level of involvement, interest, and influence on the KT process [27]. Stakeholders in KT projects in community health are multidisciplinary, including decision makers in institutions providing healthcare in the community—specifically primary care—supportive services affecting social determinants of health, such as education, healthcare charities, municipal services, the media, and influencers relevant to the local community; and the local private sector involved with community health, such as private hospitals and pharmacies. Finally, community representation may include several methods, such as the use of community health needs assessments. The early integration of the community is likely to enhance overall engagement with interventions aimed at improving their health and wellbeing [28].

Effective integration will facilitate the development of realistic, community-oriented goals and outcomes for the KT process. Specific, measurable, achievable, relevant, and time-bound goals are those that are specific to a particular health issue set as a priority, measurable via specific indicators to enable evaluation, achievable according to the local context’s capacities, relevant to the local community, and time-bound according to the KT roadmap. The goals and outcomes can involve more than one translational phase, such as dissemination, implementation, and sustainability, in addition to those related to the overall KT process.

The mitigation of facilitators and challenges from a contextual perspective allows the early identification of community-related social, economic, and environmental determinants, as well as healthcare capacity factors, that may act as either facilitators or challenges to the implementation of interventions and the overall KT process. This stage of the KT process is important in ensuring the optimal utilization of facilitators and targeted efforts to overcome potential challenges. Facilitators of KT in community health may include the suitability of the utilized knowledge to the target community, a supportive culture, motivated knowledge users, effective community communication, and adequate access to primary and community health services. Challenges may include the limited availability of cost-effective community interventions, cultural conflicts, limited organizational support, or constrained resources.

Knowledge translation frameworks in the community health and primary care contexts may benefit from involving multiple frameworks pertaining to both community health and KT. For example, the utilization of CBPR can enable the active engagement of community members in setting goals related to evidence synthesis and enhance overall engagement and sustainability [29]. The SEM, which involves multiple levels of community engagement, can aid in the integration process by targeting the individual, relationship, community, and societal levels [30]. Finally, frameworks related to KT within a community health context, such as the KTA framework, the Reach, Effectiveness, Adoption, Implementation, Maintenance (RE-AIM) framework, and the Practical Implementation Sustainability Model, can be used to integrate factors and findings gained from CBPR and the SEM to guide the cycle of translating knowledge into sustainable actions [13,31].

Evidence in the community health and primary care contexts can be a combination of data obtained from the local community, the utilization of primary care services, and the relevant health literature. For example, secondary data gathered from the healthcare system or independent investigations conducted within the same community can provide significant insights into health priorities at the community level and help to determine whether a planned intervention is suitable [32]. Additionally, perspectives gained from relevant stakeholders, professionals, and community members through community health needs assessments can further enrich the knowledge created and enhance its translation by linking the findings to the local context. Finally, the relevant health literature suggesting the best evidence-based interventions for similar health priorities can support the generation of interventions that are both suitable for the community and applicable through primary care services in collaboration with other community institutions. It should also be noted that the cyclic completion of KT can further advance knowledge and support better planning for future community health projects.

Goals, outcomes, and the created knowledge should be effectively mobilized and disseminated to enable successful KT. Knowledge mobilization is a broader term that can include knowledge dissemination [33,34]. The mobilization of knowledge in community health has important characteristics, as it ensures targeted, multidirectional, proactive, and shared-ownership communication to guarantee the availability and usefulness of the synthesized knowledge to potential knowledge users. This differs from the diffusion of knowledge, which is passive, unplanned, and uncontrolled and usually occurs when users seek the knowledge themselves [35]. Therefore, a dissemination plan should be an integral part of the KT process, with early initiation, appropriate message development according to the audience type, the selection of suitable dissemination methods, dedicated knowledge brokers [36], and clear dissemination evaluation indicators, such as accessibility and impact, budgeting, and sustainability.

Previous steps of the KT should ensure that the mobilized community health interventions are implemented with clear and practical implementation goals. This involves having a detailed roadmap of the implementation, shared with the knowledge users in the primary care context, with clear indicators to assess the impact and suitability of the implementation for the community. Assessment of the implementation impact should be transparent and involve composing a clear protocol that covers aspects related to the early selection of indicators, the decision regarding who will perform the evaluation, the required budgeting and timeline, the methods of quantitative and qualitative measurement, and the analysis plan.

Communicating the findings of the evaluation and the resulting evidence concerning effectiveness and suitability to the community can enhance implementation by addressing relevant challenges and supporting the use of facilitators. This can improve the performance of the applied intervention and modify it if necessary. The success of the intervention may facilitate its incorporation into routine practice, its adoption by primary care services through the education and training of primary healthcare workers, and subsequent support for the sustainability of the intervention and the overall success of the KT process.

### 3.3. Situational Assessment of Health Research and Evidence-Based Practice in Saudi Arabia

Saudi Arabia fosters a supportive research environment, especially in health and wellness research. According to the Saudi Research Development and Innovation Authority, the research output in Saudi Arabia increased from 15,000 publications in 2014 to 56,000 publications in 2023, with a growth rate of 16%, making Saudi Arabia one of the leading countries in research growth globally. Additionally, health and wellness research in Saudi Arabia represents 20% of the country’s research output growth [37]. This indicates the significant awareness among Saudi Arabian healthcare establishments of the importance of research in healthcare and the promotion of evidence-based practice to enhance healthcare delivery quality and effectiveness.

Currently, health research and evidence-based practice are supported by multiple higher education institutions, healthcare institutions, and governmental research agencies (Figure 2). One of the leading institutions supporting health research in Saudi Arabia is the Saudi National Institute of Health (Saudi NIH). The Saudi NIH is an initiative of the Saudi Ministry of Health and represents an implementation of one of the realization programs of Saudi Vision 2030. The institute was established in 2023 by a decision of the Saudi Council of Ministers. The Saudi NIH has multiple strategic goals, including funding translational research and transforming health research outcomes. The KT department within the Saudi NIH fosters an environment supporting health research translation via three main core functions, namely knowledge creation, knowledge mobilization, and knowledge implementation [38].

Other healthcare providers supporting health research in Saudi Arabia include the King Faisal Specialist Hospital and Research Center, which is one of the leading tertiary healthcare facilities and research establishments in the country [39]. The King Abdullah International Medical Research Center, affiliated with the Saudi Ministry of National Guard, is another important research center in Saudi Arabia [40]. The Saudi Council of Health is a strategic institution involved in healthcare strategy planning in the country, including the establishment of the National Center for Evidence-Based Medicine, with the overall objective of sustaining evidence-based practice in healthcare [41].

Other institutions and agencies involved in either regulating or funding health research in Saudi Arabia include the Research Development and Innovation Authority [42], the King Abdulaziz City for Science and Technology [43], and the Saudi Food and Drug Authority [44]. Leading Saudi universities host multiple research centers and research chairs, many of which are related to health specialties, such as King Saud University [45] and King Abdulaziz University [46]. Additionally, the National Center for the Non-Profit Sector contributes to supporting research in certain institutions, such as the King Salman Center for Disability Research [47]. This growing number of establishments involved in regulating, funding, and conducting research suggests increasing interest in promoting health research in Saudi Arabia. Nonetheless, this may complicate KT due to possible variability in the strategic goals of these agencies.

### 3.4. Perspectives on Knowledge Translation in Community Health and Primary Care Under the New Saudi Model of Care

The Saudi Model of Care emphasizes the importance of healthy communities and primary care as early elements of healthcare service delivery before reaching the secondary and tertiary healthcare levels [48]. Additionally, healthcare transformation under Saudi Vision 2030 encourages the utilization of evidence to improve healthcare services provided to individuals and enhance the standardization of healthcare delivery [12]. Nonetheless, the generation of evidence, the proper communication and dissemination of this evidence, and its application and utilization in community health and primary care settings are likely to be complex and require dynamic interactions compared to the application of evidence-based practice at higher levels of healthcare.

The new Saudi Model of Care is based on three main pillars: mental health, social health, and physical health [49]. One of the components of the Saudi Model of Care is the “Keep Well” initiative, which includes multiple programs such as the Health Coach Program, Community Health Promotion Program, Workplace Health Promotion Program, School Health Promotion Program, Healthy Eating Program, Education through Entertainment Program, and Population Health Promotion. Health Holding, one of the leading healthcare service providers in Saudi Arabia, associated with the realization of the healthcare transformation program, promotes integrative healthcare by securing effective collaboration with all community institutions and enhancing individuals’ healthy interactions within healthy communities [50].

It can be noted that several components involved in the new Saudi Model of Care are directed toward the promotion of health in community settings. The success of the new Saudi Model of Care necessitates the adoption and application of KT principles in community health and primary care based on the following notions. The roadmap for planning KT in community health and primary care, illustrated in Figure 1, necessitates early integration between the KT team and knowledge users. In the context of the new Saudi Model of Care, Health Holding represents one of the main knowledge users in the healthcare domain in Saudi Arabia, while the institutions illustrated in Figure 2, such as the SNIH, and universities, represent an important component of the KT team. Therefore, the early integration of these institutions is necessary to allow the effective implementation of the remaining steps of the KT roadmap illustrated in Figure 1.

Integration between the KT team and knowledge users is important to generate community-oriented goals and outcomes that are adopted and agreed upon by the end users, such as healthcare workers. This concept can be challenged if healthcare workers are not fully involved, especially with the ongoing healthcare transformation under the New Model of Care. Suleiman and Ming indicated that healthcare workers in Saudi Arabia might experience hesitancy concerning the adoption of the New Model of Care due to fear of the unknown [12]. Furthermore, Chowdhury et al. indicated that the new Saudi Model of Care focuses on enhancing the awareness of the community and establishing effective collaboration between organizations involved with the provision of healthcare in the country [49]. This signifies the importance of early integration, shared goal settings, and the careful assessment of facilitators and challenges that could hinder the application of KT within the new Saudi Model of Care, especially with the presence of a diversity of sectors involved with the overall community health.

The new Saudi Model of Care adopts concepts related to evidence synthesis, the development of guidelines, the mobilization of the guidelines, and implementation within healthcare settings [5,12]. This practice is likely to enhance the provision of standardized care, and, therefore, enable the assessment of its effectiveness through the evaluation process. According to Al Trad et al., the new Saudi Model of Care involves 42 initiatives that are involved in primary care within some phases. These initiatives have specific pathways and key performance indicators to assess the outcomes, with the overall aim of providing knowledge to individuals as part of the process of improving their health [48]. This indicates the complexity of the evaluation process, and the careful application of KT principles is necessary to ensure sustainability.

## 4. Discussion

This narrative review provides a conceptual overview of KT and its application in community health and primary care contexts. Additionally, the review illustrates how to apply the principles of KT during the establishment of the new Saudi Model of Care within primary healthcare and community health services. KT provides a roadmap for successful connection between research and real-world healthcare practice. KT is different from research-driven practices, which may lack direct application in a healthcare context. Therefore, the application of KT is crucial in community health and primary care contexts, where it ensures a collaborative, multidisciplinary process to facilitate the implementation and sustainability of interventions directed toward the community.

The new Saudi Model of Care emphasizes the importance of improving community health and enhancing accessibility to primary care to reduce the overutilization of secondary and tertiary healthcare services. Additionally, the model encourages collaboration and integration between research bodies, healthcare providers, and community institutions to ensure meeting the health needs of the community while applying evidence-based approaches to measure the effectiveness of applied interventions. Based on these findings, integrating KT principles within the community health and primary care contexts is of the utmost importance for the success of initiatives applied within the new Saudi Model of Care.

The findings of the review indicate that the evidence for the use of KT in community health and primary care under the new Saudi Model of Care is currently limited. This suggests that the adoption of KT principles in community health and primary care is limited. The findings of this narrative review indicate the diversity of institutions involved in the generation of knowledge associated with healthcare, institutions associated with the delivery of healthcare, and institutions involved in the overall health determinants of communities. This signifies the importance of unifying efforts to enhance the adoption of KT while acknowledging this diversity for the successful integration of the overall KT process.

The findings of the current review are supported by efforts performed previously in Saudi Arabia to enhance community health. In 2008, the Saudi Ministry of Health established a local community health program called the Crown Health Project in the Al-Jouf region, in the north of Saudi Arabia. The project aimed to evaluate the suitability and effectiveness of an implementation to prevent chronic non-communicable diseases. Memish et al. reported that community members, key regional leaders, and healthcare workers were involved during the implementation process. Nonetheless, the implementation ceased due to funding-related issues. Additionally, it was indicated that establishing connections with the local media and tribal leaders and keeping them updated on the progress of the implementation was necessary to ensure its success. Finally, Memish et al. highlighted the critical importance of providing trained healthcare personnel to participate in delivering the intervention to the community [51].

This project, implemented prior to the launch of the Saudi Model of Care, serves as a case study highlighting the importance of early planning for KT, proactive knowledge mobilization, and implementation planning to ensure sustainability. The literature assessing the implementation of community health programs in Saudi Arabia is limited to studies evaluating community health needs in certain regions of the country [52,53,54,55,56]. Nonetheless, the utilization of evidence-based practice in other clinical settings can provide insights into KT in healthcare within the country.

In a study involving 284 primary healthcare nurses from Riyadh, Saudi Arabia, the participants’ views on barriers to implementing evidence-based practice were assessed. Factors such as limited evidence specific to nursing settings, inadequate facilities supporting evidence-based practice, and limited cooperation from other healthcare workers, including physicians, were identified as the main barriers influencing the implementation of evidence-based practice in primary care [57]. In another study involving 288 physicians from the Qassim region that aimed to assess their awareness and utilization of evidence-based practice, it was concluded that training physicians to understand the principles of evidence-based practice was necessary to enhance the adoption of evidence relevant to their practice [58]. Challenges related to applying KT in other healthcare settings are also important considerations for the community health and primary care contexts in Saudi Arabia.

The current narrative review has multiple strengths and limitations. Its strength stems from the conceptual assessment of KT principles to enable its adoption in community health and primary care settings under the new Saudi Model of Care. The utilization of a narrative review as a methodology allowed a wide search of the relevant literature to illustrate the principles of KT and how it intertwines with community health and primary healthcare, provide an overview of research in the healthcare context in Saudi Arabia, and highlight the importance of applying KT principles within the new Saudi Model of Care. Nonetheless, the main limitation of the current study is due to the inability to utilize a systematic review methodology due to the large scope of the current review’s aims. Additionally, the limited evidence concerning the utilization of KT principles within the Saudi Arabian healthcare system, especially in community health and primary healthcare, may have impacted the overall synthesis of the findings.

## 5. Conclusions

This review underscores the centrality of KT in enhancing community health and primary care within Saudi Arabia’s evolving healthcare system. The establishment of the new Saudi Model of Care provides an approach that supports KT within primary healthcare and community health services. While the country has achieved significant advances in health research and institutional capacity building, KT remains underutilized in local community health initiatives. Key gaps include insufficient collaboration between researchers and knowledge users, limited awareness of KT, and challenges in sustaining evidence-informed programs. For KT to be impactful in community health and primary care, it must be embedded throughout all phases—starting with the early engagement of stakeholders, continuing with the integration of local contexts and cultural values, and culminating in the application of multidisciplinary interventions that are effective and sustainable. Future directions should focus on expanding KT training in the country, promoting its implementation across health institutions involved in primary care and community health, and fostering partnerships among academic, governmental, and community entities.

## Figures and Tables

**Figure 1 healthcare-13-02469-f001:**
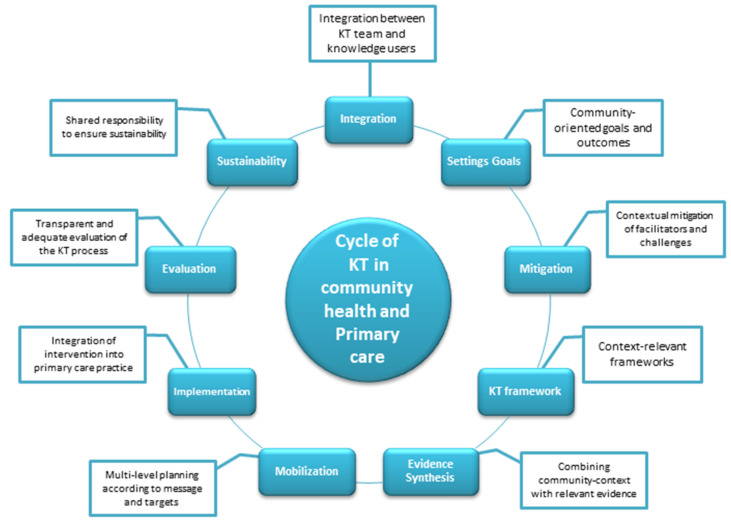
Roadmap for planning of knowledge translation (KT) in a community health and primary care context.

**Figure 2 healthcare-13-02469-f002:**
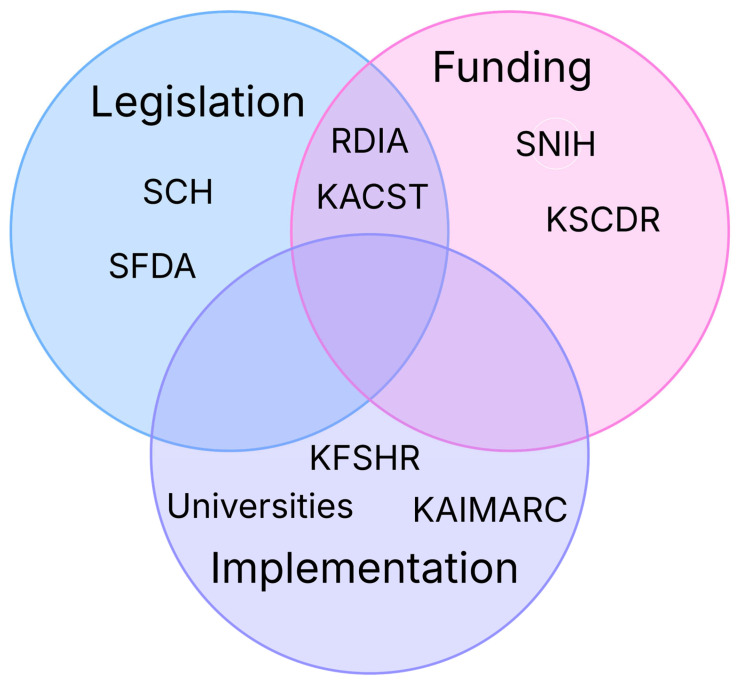
Institutions supporting health research in Saudi Arabia according to their roles in legislative, funding, or research implementation. SCH: Saudi Council of Health, SFDA: Saudi Food and Drug Authority, RDIA: Research Development and Innovation Authority, KSCST: King Abdulaziz City for Science and Technology, SNIH: Saudi National Institute of Health, KSCDR: King Salman Center for Disability Research, KFSHR: King Faisal Specialist Hospital and Research Center, KAIMARC: King Abdullah International Medical Research Center.

## Data Availability

The data presented in this work are available in the public domain.

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
