# Peer review of "Knowledge Translation of Healthcare Research in Saudi Arabia—Implications for Community Health and Primary Care Under the New Saudi Model of Care: A Narrative Review"

_healthcare, 2025, doi:10.3390/healthcare13192469_

Round 1
Reviewer 1 Report
Comments and Suggestions for Authors
The introduction is incomplete.
The necessity of doing the work and the gap are not clear.
First, relevant literature explaining the principles of KT was consulted. This was followed by a review of literature concerning community health and primary care and how these
areas intersect with KT principles. Second, an overview was developed to present the
current status of KT in Saudi Arabia, identifying the main institutions responsible for
legislating and implementing knowledge creation and translation, by searching all rele
vant research-performing entities in the country. Third, studies assessing evidence-based
practice and KT in community health and primary care in Saudi Arabia were reviewed.
These steps were carried out by searching PubMed and Google Scholar using terms re
lated to KT, evidence-based practice, Saudi Model of Care, community health, and pri-....
It's very confusing.
The methodology is very incomplete.
What is the difference between case 1 and 3?
For case 1 and 3, proper search, evaluation of articles, etc. should be clearly stated.
For case 2, a detailed qualitative study is also needed.
In general, the methodology is incomplete and inaccurate, so the results are not reliable and appropriate.
The results are presented in a boring manner, without any tables.
Many parts of the results are long and lack references.
The conclusion is long and not based on the results.
Author Response
Responses to Reviewer 1 Comments
Comment: The introduction is incomplete.
The necessity of doing the work and the gap are not clear.
Response: A new section is now added to explain nature and development of the new Saudi Model of Care. A description of the components of the model, and how it is designed to involve multiple institutions in the community is made. Additionally, more emphasis is made concerning why knowledge translation is critically important for successful application of the Saudi model of care since it promotes integrative healthcare by securing effective collaboration with all community institutions and enhancing individuals’ healthy interactions within healthy communities.
The following was added to the introduction to illustrate this point:
‘In Saudi Arabia, KT has gained increasing importance as the healthcare system undergoes a major transformation under Saudi Vision 2030. The Vision provides a comprehensive roadmap leading to a revolutionary transformation of healthcare in Saudi Arabia. The Health Sector Transformation Program is one of the vision realization programs, aiming to enhance access to healthcare in a manner that ensures meeting the health needs of every individual in the community [5]. The program emphasizes the importance of public health and disease prevention to ensure the health and productivity of individuals and communities.
One of the main health transformation themes introduced by the Saudi Ministry of Health is the New Model of Care. The newly introduced model, recently re-labeled as the Saudi Model of Care, focuses on enhancing personal value during the healthcare-seeking process [6]. Earlier reports indicated that accessibility to healthcare in Saudi Arabia was facing challenges , especially with the limited impact of primary healthcare services and growing burden of chronic diseases [7]. The transformation of healthcare system under the Saudi Vision 2030 emphasized on the importance of responding to the health needs of the population and adoption of preventive approach, instead of the previously adopted curative approach [8]. Additionally, the transformation included a shift from independent healthcare services provision toward integrated healthcare services across all levels of healthcare services, including private healthcare services, while adopting use of electronic healthcare systems [9, 10].’
Comment: First, relevant literature explaining the principles of KT was consulted. This was followed by a review of literature concerning community health and primary care and how these areas intersect with KT principles. Second, an overview was developed to present the current status of KT in Saudi Arabia, identifying the main institutions responsible for legislating and implementing knowledge creation and translation, by searching all relevant research-performing entities in the country. Third, studies assessing evidence-based practice and KT in community health and primary care in Saudi Arabia were reviewed. These steps were carried out by searching PubMed and Google Scholar using terms related to KT, evidence-based practice, Saudi Model of Care, community health, and pri-....
It's very confusing.
Response:
Benefiting from your assessment and review, the manuscript has updated the manuscript with a comprehensive revision including clear identification of knowledge gap necessitating conducting the review, provision of specific and clear aims, rigorous, transparent and detailed literature search strategy, clear outline of the findings of the review to minimize confusion.
Comment: The methodology is very incomplete.
Response:
Major revision is now applied to the Aims and Methodology section. This involved clear description of the knowledge gap in the first section, followed up by steps taken to perform the review, targeted databases, utilized terminology, search strategy, inclusion and exclusion criteria, and nature of identified literature. Modifications applied to this section are detailed as the following:
‘To conduct this review, multiple steps were taken to gather content related to the current status of KT in Saudi Arabia and its connection to community health and primary care. First, relevant literature explaining the principles of KT was consulted. This was followed by a review of literature concerning community health and primary care and how these areas intersect with KT principles. Second, an overview was developed to present the current status of KT in Saudi Arabia, identifying the main institutions responsible for legislating and implementing knowledge creation and translation, by searching all relevant research-performing entities in the country. Third, studies assessing evidence-based practice and KT in community health and primary care in Saudi Arabia were reviewed.
These steps were carried out by searching Web of Science, PubMed, Google Scholar, and region-specific literature searching databases such as the Saudi Digital Library. The utilized keywords and MeSH terms were related to knowledge translation, evidence-based practice, Saudi Model of Care, community health, and primary care. This involved use of specific terms and key words related to these main concepts such as community care, community health services, and community medicine, primary health care, access to primary care, knowledge translation, translational medical sciences, evidence-based practice, evidence-based healthcare, and healthcare reform.
To enhance the ability to identify literature relevant to the review aims, the advanced search strategy to identify published peer-reviewed articles involved the use of Boolean operators (such as AND, OR), inclusion of the selected terms in the title or abstract of identified articles. No time frame restriction was applied for the searched literature. Additionally, grey literature search involved targeting recognized resources and authorities associated with the KT in healthcare, KT within community and primary care, evidence-based practice in healthcare settings in Saudi Arabia, and the new Saudi Model of Care. Finally, hand searching of reference list of included articles was utilized to identify potentially other relevant literature.
The inclusion and exclusion criteria of the identified literature were based on the relevance to KT and its application within community and primary healthcare context. Articles were included if they are published in English language, presenting original research, systematic or narrative reviews, statements from recognized establishments and official institutions pertaining to KT, evidence-based practice, Saudi Model of Care, community health, and primary care. Exclusion of identified literature was performed if the topics were not relevant to KT within community healthcare and primary healthcare or if not related to the transformation of healthcare within Saudi Arabian context.’
Comment: What is the difference between case 1 and 3?
For case 1 and 3, proper search, evaluation of articles, etc. should be clearly stated.
For case 2, a detailed qualitative study is also needed.
Response:
While this comment is not entirely clear for this reviewer as the manuscript did not involve presentation of cases, an extensive effort was made to enhance the writing quality of the manuscript.
Comment: In general, the methodology is incomplete and inaccurate, so the results are not reliable and appropriate.
Response:
Major revision is now applied to the Aims and Methodology section. This involved clear description of the knowledge gap in the first section, followed up by steps taken to perform the review, targeted databases, utilized terminology, search strategy, inclusion and exclusion criteria, and nature of identified literature. Modifications applied to this section are detailed as the following:
‘To conduct this review, multiple steps were taken to gather content related to the current status of KT in Saudi Arabia and its connection to community health and primary care. First, relevant literature explaining the principles of KT was consulted. This was followed by a review of literature concerning community health and primary care and how these areas intersect with KT principles. Second, an overview was developed to present the current status of KT in Saudi Arabia, identifying the main institutions responsible for legislating and implementing knowledge creation and translation, by searching all relevant research-performing entities in the country. Third, studies assessing evidence-based practice and KT in community health and primary care in Saudi Arabia were reviewed.
These steps were carried out by searching Web of Science, PubMed, Google Scholar, and region-specific literature searching databases such as the Saudi Digital Library. The utilized keywords and MeSH terms were related to knowledge translation, evidence-based practice, Saudi Model of Care, community health, and primary care. This involved use of specific terms and key words related to these main concepts such as community care, community health services, and community medicine, primary health care, access to primary care, knowledge translation, translational medical sciences, evidence-based practice, evidence-based healthcare, and healthcare reform.
To enhance the ability to identify literature relevant to the review aims, the advanced search strategy to identify published peer-reviewed articles involved the use of Boolean operators (such as AND, OR), inclusion of the selected terms in the title or abstract of identified articles. No time frame restriction was applied for the searched literature. Additionally, grey literature search involved targeting recognized resources and authorities associated with the KT in healthcare, KT within community and primary care, evidence-based practice in healthcare settings in Saudi Arabia, and the new Saudi Model of Care. Finally, hand searching of reference list of included articles was utilized to identify potentially other relevant literature.
The inclusion and exclusion criteria of the identified literature were based on the relevance to KT and its application within community and primary healthcare context. Articles were included if they are published in English language, presenting original research, systematic or narrative reviews, statements from recognized establishments and official institutions pertaining to KT, evidence-based practice, Saudi Model of Care, community health, and primary care. Exclusion of identified literature was performed if the topics were not relevant to KT within community healthcare and primary healthcare or if not related to the transformation of healthcare within Saudi Arabian context.’
Comment: The results are presented in a boring manner, without any tables.
Many parts of the results are long and lack references.
Response:
An effort was made to enhance the presentation of the results. However, since this review is narrative, two figures were used to present a visual summary of the findings instead of using tables.
Comment: The conclusion is long and not based on the results.
Response: A new section is now added to discuss the findings of the review. The conclusion was written to ensure provision of a summary of the main findings and practical recommendation based on the presented findings.
Reviewer 2 Report
Comments and Suggestions for Authors
The manuscript presents an interesting and valuable contribution to the field. However, the methodology, as currently described, lacks clarity and rigor. At present, it does not conform to established standards for either a systematic review or an original research study.
Without a transparent, systematic methodology and clear visual summaries, the paper rigor remains in question. A full systematic review presentation would markedly improve the manuscript’s quality and scholarly impact.
To enhance the manuscript’s methodological strength and credibility, I recommend revising it to fully embrace the framework of a systematic review. Specifically, the authors should:
- Clearly articulate the search strategy, detailing databases and sources used (e.g., PubMed, Scopus, Web of Science); specific search terms, keywords, and any filters or limits applied; date ranges covered; language restrictions, if any.
- Explicitly define the eligibility criteria for study inclusion and exclusion, such as study design, population/sample characteristics, outcomes measured, publication type.
- Describe the study selection process, including: how many reviewers screened titles/abstracts and full-texts, whether disagreements were resolved by consensus or via a third reviewer.
- Include details about data extraction (what information was extracted from each study, how data consistency was ensured).
To further improve readability and clarity, the authors would benefit for presenting the following items visually, for example with a PRISMA-style flow diagram summarizing the number of records identified, screened, excluded (with reasons), and ultimately included in the review, and with summary tables (characteristics of included studies-study authors, year, design, sample size, main findings, a synthesis matrix or comparative table highlighting key outcomes and patterns across studies…).
Author Response
Responses to Reviewer 2 Comments
Comments: The manuscript presents an interesting and valuable contribution to the field. However, the methodology, as currently described, lacks clarity and rigor. At present, it does not conform to established standards for either a systematic review or an original research study.
Without a transparent, systematic methodology and clear visual summaries, the paper rigor remains in question. A full systematic review presentation would markedly improve the manuscript’s quality and scholarly impact.
To enhance the manuscript’s methodological strength and credibility, I recommend revising it to fully embrace the framework of a systematic review. Specifically, the authors should:
- Clearly articulate the search strategy, detailing databases and sources used (e.g., PubMed, Scopus, Web of Science); specific search terms, keywords, and any filters or limits applied; date ranges covered; language restrictions, if any.
- Explicitly define the eligibility criteria for study inclusion and exclusion, such as study design, population/sample characteristics, outcomes measured, publication type.
- Describe the study selection process, including: how many reviewers screened titles/abstracts and full-texts, whether disagreements were resolved by consensus or via a third reviewer.
- Include details about data extraction (what information was extracted from each study, how data consistency was ensured).
To further improve readability and clarity, the authors would benefit for presenting the following items visually, for example with a PRISMA-style flow diagram summarizing the number of records identified, screened, excluded (with reasons), and ultimately included in the review, and with summary tables (characteristics of included studies-study authors, year, design, sample size, main findings, a synthesis matrix or comparative table highlighting key outcomes and patterns across studies…).
Response:
It is now clearly stated that this paper is a narrative review, either early in the title, or within the introduction. Additionally, more effort is now made to outline a logical and reproducible strategy for literature search within the methodology section. The following was added to enhance the reporting of the utilized literature search strategy:
‘To conduct this review, multiple steps were taken to gather content related to the current status of KT in Saudi Arabia and its connection to community health and primary care. First, relevant literature explaining the principles of KT was consulted. This was followed by a review of literature concerning community health and primary care and how these areas intersect with KT principles. Second, an overview was developed to present the current status of KT in Saudi Arabia, identifying the main institutions responsible for legislating and implementing knowledge creation and translation, by searching all relevant research-performing entities in the country. Third, studies assessing evidence-based practice and KT in community health and primary care in Saudi Arabia were reviewed.
These steps were carried out by searching Web of Science, PubMed, Google Scholar, and region-specific literature searching databases such as the Saudi Digital Library. The utilized keywords and MeSH terms were related to knowledge translation, evidence-based practice, Saudi Model of Care, community health, and primary care. This involved use of specific terms and key words related to these main concepts such as community care, community health services, and community medicine, primary health care, access to primary care, knowledge translation, translational medical sciences, evidence-based practice, evidence-based healthcare, and healthcare reform.
To enhance the ability to identify literature relevant to the review aims, the advanced search strategy to identify published peer-reviewed articles involved the use of Boolean operators (such as AND, OR), inclusion of the selected terms in the title or abstract of identified articles. No time frame restriction was applied for the searched literature. Additionally, grey literature search involved targeting recognized resources and authorities associated with the KT in healthcare, KT within community and primary care, evidence-based practice in healthcare settings in Saudi Arabia, and the new Saudi Model of Care. Finally, hand searching of reference list of included articles was utilized to identify potentially other relevant literature.
The inclusion and exclusion criteria of the identified literature were based on the relevance to KT and its application within community and primary healthcare context. Articles were included if they are published in English language, presenting original research, systematic or narrative reviews, statements from recognized establishments and official institutions pertaining to KT, evidence-based practice, Saudi Model of Care, community health, and primary care. Exclusion of identified literature was performed if the topics were not relevant to KT within community healthcare and primary healthcare or if not related to the transformation of healthcare within Saudi Arabian context.’
Finally, since this paper is a narrative review, many of the suggestions are more associated with systematic reviews and, therefore, are not applicable on the current submission.
Reviewer 3 Report
Comments and Suggestions for Authors
The topic is interesting, as is the study's purpose.
It proposes a literature review that seeks to compare the findings with the reality in Saudi Arabia.
However, the method doesn't clearly state which databases were used, nor the criteria for including or excluding materials in the study. Therefore, it needs to describe the step-by-step process of study selection, such as:
1. Which databases were used?
2. What languages were included in the sample?
3. Was there a time limit?
4. Were paid studies used or only free ones?
5. Was the selection performed in pairs?
6. Did you use any application to sort the material or to assist in analyzing the material?
7. What are the keywords and/or descriptors used to select the material?
8. What is the data collection period?
All of these questions must be described in the method.
It is also recommended that a table with the bases and other data for the inclusion and/or exclusion of materials be included.
Most of the references used are more than 10 years old, it is recommended to update and leave the majority with a maximum of 5 years.
Author Response
Responses to Reviewer 3 Comments
Comment: The topic is interesting, as is the study's purpose.
It proposes a literature review that seeks to compare the findings with the reality in Saudi Arabia.
Response: The author of the manuscript appreciate the supportive comment of the reviewer. A comprehensive revision is now made to enhance the writing quality of the manuscript.
Comment: However, the method doesn't clearly state which databases were used, nor the criteria for including or excluding materials in the study. Therefore, it needs to describe the step-by-step process of study selection, such as:
- Which databases were used?
- What languages were included in the sample?
- Was there a time limit?
- Were paid studies used or only free ones?
- Was the selection performed in pairs?
- Did you use any application to sort the material or to assist in analyzing the material?
- What are the keywords and/or descriptors used to select the material?
- What is the data collection period?
All of these questions must be described in the method.
It is also recommended that a table with the bases and other data for the inclusion and/or exclusion of materials be included.
Response:
It is now clearly stated that this paper is a narrative review, either early in the title, or within the introduction. Additionally, more effort is now made to outline a logical and reproducible strategy for literature search within the methodology section. The following was added to enhance the reporting of the utilized literature search strategy:
‘To conduct this review, multiple steps were taken to gather content related to the current status of KT in Saudi Arabia and its connection to community health and primary care. First, relevant literature explaining the principles of KT was consulted. This was followed by a review of literature concerning community health and primary care and how these areas intersect with KT principles. Second, an overview was developed to present the current status of KT in Saudi Arabia, identifying the main institutions responsible for legislating and implementing knowledge creation and translation, by searching all relevant research-performing entities in the country. Third, studies assessing evidence-based practice and KT in community health and primary care in Saudi Arabia were reviewed.
These steps were carried out by searching Web of Science, PubMed, Google Scholar, and region-specific literature searching databases such as the Saudi Digital Library. The utilized keywords and MeSH terms were related to knowledge translation, evidence-based practice, Saudi Model of Care, community health, and primary care. This involved use of specific terms and key words related to these main concepts such as community care, community health services, and community medicine, primary health care, access to primary care, knowledge translation, translational medical sciences, evidence-based practice, evidence-based healthcare, and healthcare reform.
To enhance the ability to identify literature relevant to the review aims, the advanced search strategy to identify published peer-reviewed articles involved the use of Boolean operators (such as AND, OR), inclusion of the selected terms in the title or abstract of identified articles. No time frame restriction was applied for the searched literature. Additionally, grey literature search involved targeting recognized resources and authorities associated with the KT in healthcare, KT within community and primary care, evidence-based practice in healthcare settings in Saudi Arabia, and the new Saudi Model of Care. Finally, hand searching of reference list of included articles was utilized to identify potentially other relevant literature.
The inclusion and exclusion criteria of the identified literature were based on the relevance to KT and its application within community and primary healthcare context. Articles were included if they are published in English language, presenting original research, systematic or narrative reviews, statements from recognized establishments and official institutions pertaining to KT, evidence-based practice, Saudi Model of Care, community health, and primary care. Exclusion of identified literature was performed if the topics were not relevant to KT within community healthcare and primary healthcare or if not related to the transformation of healthcare within Saudi Arabian context.’
Comment: Most of the references used are more than 10 years old, it is recommended to update and leave the majority with a maximum of 5 years.
Response: An effort was made to add more recent articles. However, this review made a historical comparison between an older model of care and a new model of care in Saudi Arabian context. Therefore adding older references was necessary.
Round 2
Reviewer 2 Report
Comments and Suggestions for Authors
The authors addressed my comments.
Author Response
Comment: The authors addressed my comments.
Response: The author of the manuscript appreciates the supportive comment of the reviewer. It is now ensured that the quality of the manuscript has improved due to revisions made as per the comments made by the reviewer.
Reviewer 3 Report
Comments and Suggestions for Authors After adjustments by the authors, I consider that the suggestions were met and the article isready for publication.
Author Response
Comment: After adjustments by the authors, I consider that the suggestions were met and the article is
ready for publication.
Response: The author of the manuscript appreciates the supportive comment of the reviewer. It is now ensured that the quality of the manuscript has improved due to revisions made as per the comments made by the reviewer.